# Effects of Ti Target Purity and Microstructure on Deposition Rate, Microstructure and Properties of Ti Films

**DOI:** 10.3390/ma15072661

**Published:** 2022-04-05

**Authors:** Liming Liu, Wuhui Li, Haoliang Sun, Guangxin Wang

**Affiliations:** 1School of Materials Science and Engineering, Henan University of Science and Technology, Luoyang 471003, China; liuliming9511@163.com (L.L.); whli@haust.edu.cn (W.L.); 2Luoyang Key Laboratory of High Purity Materials and Sputtering Targets, Henan University of Science and Technology, Luoyang 471003, China

**Keywords:** Ti target, purity, microstructure, deposition rate, resistivity

## Abstract

Three titanium (Ti) targets with different purities were used to prepare Ti films on polyimide substrates by DC magnetron sputtering. The microstructures of Ti films were characterized by a metallographic microscope, X-ray diffractometer, field emission scanning electron microscope and three-dimensional surface topography instrument. In this study, we investigated the effects of Ti target purity and microstructure on film deposition rate, surface roughness, microstructure and resistivity. The results show that the deposition rate increased with increasing Ti target purity. Ti film deposited by the high-purity (99.999%) Ti target has fewer surface particles with smaller size, lower surface roughness and lower resistivity when compared to that prepared by the Ti target of low purity (99.7%). The surface roughness of Ti film prepared by the high-purity Ti target was Sa = 121 nm, the deposition rate was 16.3 nm/min and the resistivity was 6.9 × 10^−6^ Ω·m. For Ti targets of the same purity, the performance of Ti film prepared by a target with equiaxed α-phase grains is better than that of Ti film prepared by a target with twins and β-phase grains.

## 1. Introduction

Because of its high wear and corrosion resistance, biocompatibility and good electrical conductivity, titanium (Ti) is widely used in aerospace, biomedicine, decorative coatings and other fields [1,2,3]. At the same time, high-purity Ti films also have important applications in integrated circuits and the semiconductor industry [4]. As we know, the Ti target is an indispensable raw material in preparing Ti film by magnetron sputtering and is closely related to the performance of Ti film [5]. As an important part of electronic materials, high-purity metal sputtering targets have penetrated into all fields of the electronic information industry [6]. With the increasing integration of microelectronic devices, application conditions of thin solid films are becoming more demanding, and the expectations for the performance of films used to prepare devices are also becoming higher [7]. Therefore, there are higher [8] quality requirements for the purity, grain size and density of the target. Higher content of impurity elements in the Ti target can promote the formation of convex particles on the surface of the films [9]. These particles can cause short circuits or open circuits in interconnection cords, which can seriously affect the service life and safety of microdevices. Therefore, the properties of films can be improved by increasing target purity to reduce contamination sources for deposited films. Studies have shown that [10] the properties of Ti film are mainly related to the purity, grain size, structure uniformity and crystal orientation of the Ti target under the same sputtering process parameters.

Previous studies on the sputtering properties of Ti targets mainly focused on obtaining pure Ti films and Ti alloy films with better performance by changing the sputtering process parameters. Bingxun [11] prepared pure titanium films by DC magnetron sputtering and found that the deposition rate increases with the increase in sputtering power. Aznilinda [12] researched the influence of sputtering power on the properties of Ti films and found that Ti films grew into a columnar structure, the film thickness and surface roughness increased with the increase in sputtering power, and the particle size gradually enlarged. Chen [13] studied the preparation of Ti films with different deposition rates on Si (100) substrate. When the deposition rate increased from 0.05 nm/s to 1.46 nm/s, the growth process of Ti films changed from two-dimensional bulk to three-dimensional island, then to two-dimensional flake, and the crystal orientation changed from random orientation to (002) preferred orientation. Previous research has mainly focused on analyzing the influence of Ti target microstructure and sputtering process parameters on the properties of Ti films, but there are very few reports on the influence of target purity on the deposition rate, microstructure and properties of Ti films. Polyimide (PI) films have good environmental stability and mechanical properties and excellent dielectric properties, and they are widely used in many basic industries and high-tech fields [14]. Therefore, this paper uses polyimide as a substrate to explore the impact of the purity and microstructure of a Ti target on the deposition, surface and cross-section morphology, surface roughness and resistivity of Ti film and to provide technical support for preparing Ti film with good performance.

## 2. Materials and Methods

A JCP-350M2 magnetron sputtering coating machine was used to prepare pure Ti thin films on flexible polyimide (PI) substrates by DC magnetron sputtering. A low-purity Ti target (99.7%), an industrial Ti target (99.99%) and a high-purity Ti target (99.999%) were used to prepare Ti thin films. Elemental analysis of Ti targets of different purities have been added to Appendix A. The annealing temperatures of the Ti targets are shown in Table 1. The size of Ti targets was Φ 50 mm × 4 mm, and the same process parameters are applied to prepare Ti films. Before sputtering, targets and PI substrates were ultrasonically cleaned with acetone, absolute ethanol and deionized water for 15 min. The distance between the target and substrate table was 70 mm. The coating process was as follows: vacuum degree was 5 × 10^−4^ Pa; the argon flow rate was 40 sccm; the sputtering power was 120 W; and the sputtering times were 10, 20, and 30 min.

Phase structures of targets A, B and C were characterized by X-ray diffraction (XRD, D8 advance, BRUKER, Ltd., Kyoto, Japan). The working parameters were as follows: the current was 40 mA, the voltage was 40 kV, the scanning speed was 6°/min and the diffraction angle was set within the range of 20–90°. The microstructure of the Ti targets was observed with a metallographic microscope (PMG3, OLYMPUS, Tykyo, Japan). The working parameters were as follows: the lamp voltage was 10 V, and the power was 20 W. Surface and cross-sectional morphologies of the films were characterized by field emission scanning electron microscopy (FESEM, JSM-7800F, JEOL Ltd., Tykyo, Japan). The working parameters were as follows: the voltage was 40 kV, and the magnification was 40,000 times. Film roughness was characterized by a three-dimensional surface topography instrument (Nanovea, model HS100P, Shanghai, China). The test parameters were as follows: the XY scanning step size was 0.1 μm, and the scanning speed was 2 mm/s. The sheet resistance of the films was measured using a four-probe resistance tester (FPP, model RTS-8). The test parameters were as follows: the sample size was 20 mm × 20 mm, and the current range was 1 mA.

## 3. Results and Discussion

### 3.1. Phase Structures of the Three Targets

Figure 1 demonstrates XRD patterns of three targets with different levels of purity. It can be seen from Figure 1 that the characteristic diffraction peaks of Ti (100), (002), (101) and (102) appear on the XRD patterns of the three Ti targets with different purities. This indicates that the three Ti targets used in this paper are all α-Ti with hexagonal close-packed structures [15]. As the purity of the Ti targets increases, the intensity of characteristic diffraction peaks gradually increases, which indicates that the higher the purity, the better the crystallization of the target under the same condition [16].

### 3.2. Microstructure of Target

Ti has two allotropes, α-Ti and β-Ti. When the temperature is below 882 °C, the microstructure of Ti is composed of α-Ti. When the temperature reaches 882 °C, α-Ti transforms into β-Ti [17]. Figure 2 shows the metallographic structures of three Ti targets with different levels of purity. It can be seen from Figure 2 that the microstructures of targets A, B and C are all composed of equiaxed α-phase grains [18] with average grain sizes of 37.3, 35.4 and 35.2 μm, respectively. The grain sizes of the three Ti targets are very similar, while the grain size of target C is more uniform. Studies have shown that a target with good structural uniformity is helpful in preparing films with more uniform thickness and fewer defects [19].

### 3.3. Surface Morphologies of Films Prepared from Ti Targets with Different Purities

Figure 3 illustrates surface morphologies of Ti films prepared by sputtering three targets on flexible polyimide (PI) substrates for 10 min with the same sputtering process parameters (sputtering power, vacuum and sputtering pressure). It can be seen from Figure 3 that surface of the Ti film prepared from target A is relatively rough due to the high number of larger particles on the surface, and particle size distribution is not uniform [20]. Figure 3B,C also shows that the surface of the Ti film prepared with targets B and C is relatively smooth, and particles formed on the surface are smaller and evenly distributed on the film surface, which is ascribed to the purity of targets and the uniformity of grain size [21].

In the case of target A with low purity, impurity elements are easily deposited into Ti film, which may lead to increased residual compressive stress in the film and promote the formation of particle protrusions, thereby affecting the surface quality of as-prepared Ti film [22,23]. As the purity of the target increases and the content of impurity elements decreases, the prepared film has more uniform grains and a smoother surface. The conclusion of this experiment is that the smoothness of sputtered Ti film surface is improved with higher Ti target purity.

Figure 4 shows the three-dimensional morphology and surface roughness of Ti films with the deposition time of 10 min. Particles on the surface of Ti films prepared by targets A and B are obviously bigger than those on Ti films prepared by target C, which is consistent with the surface morphology of the film observed by SEM and shown in Figure 3. Surface roughnesses of Ti films deposited with targets A, B and C are Sa = 170, 136 and 121 nm, respectively. The main impurities of the low-purity Ti target (99.7%) are C, H, O and Fe elements. Ti oxides in the target may hinder the diffusion of Ti atoms on the PI substrate, reduce the deposition rate and increase the surface roughness of Ti film. Results show that the surface roughness of the films gradually decreases with the increase in the purity of Ti targets, which is attributed to the larger and unevenly distributed particles on the surface of Ti films deposited by the low-purity Ti target, which leads to an increase in film surface roughness [24].

### 3.4. Cross-Sectional Morphology, Film Thickness and Resistivity of Films Prepared by Different-Purity Ti Targets

Figure 5 shows the cross-sectional morphology and thickness of Ti films prepared by sputtering for 10 min. Ti film deposited by sputtering target A has relatively uneven film thickness due to the large particles on the surface, which is consistent with the results of the Ti films’ surface morphology shown in Figure 3. Impurities and pores in the target are the main factors affecting the quality of the target and films [25]. As the purity of the Ti target increases, the thickness of sputtered Ti films gradually increases. Compared with Ti films prepared with target A, thicknesses of Ti films obtained by targets B and C increase by 2.7% and 11.2%, respectively. Impurity elements are enriched at the Ti grain boundary and can hinder the diffusion of the atoms [26]. Therefore, the higher the purity of the Ti target, the higher the deposition rate of the Ti film under the same sputtering process parameters, indicating that the purity of the Ti target is an important factor influencing the deposition rate of Ti film.

Based on the above analysis, it can be concluded that the higher the purity of the Ti target, the better the performance of Ti film. Therefore, the sputtering performance of the high-purity target C was systematically studied. Figure 6 shows the sputtering morphology of target C and the cross-sectional morphology and deposition rate of Ti films deposited by high-purity target C with different sputtering times. The particles on the surface of Ti film prepared by target C are very fine and evenly distributed on the film. Moreover, the surface of Ti film prepared by the target C has no defects and is smooth and dense. At the same time, it can be seen that the growth mode of the Ti film is columnar crystal growth [27]. The thicknesses of Ti film with sputtering times of 10, 20 and 30 min are 163.3, 265 and 310 nm, respectively, and the deposition rates are 16.3, 13.3 and 10.3 nm/min, respectively. With the extension of time, the reasons for the deposition rate decrease include two aspects. One is that it is difficult to sputter the Ti atoms inside the target. Another is that the target surface becomes enriched with larger grains with the extension of time, which result in reducing grain boundary regions. The grain boundary regions have higher energy and are more easily sputtered. So, these reasons result in a slight decrease in deposition rate with the extension of time [28]. The results show that Ti film becomes thicker and the deposition rate decreases gradually with the increase in sputtering time.

Figure 7 shows the resistivity of the films prepared by targets A, B and C with the sputtering time of 10 min. Square resistances of three Ti films measured by the four-probe resistance tester are 66.46, 55.34 and 42.31 Ω/☐, respectively. The calculation formula of film resistivity is as follows [29]:R = ρ/d(1)
where R—film square resistance, unit: Ω/☐; ρ—resistivity, unit: Ω·m; d—film thickness, unit: m.

According to Formula (1), the resistivities of Ti films prepared by targets A, B and C are 9.8 × 10^−6^, 8.3 × 10^−6^, and 6.9 × 10^−6^ Ω·m, respectively. Figure 7 reveals that the resistivity of Ti films decreases gradually as the purity of the Ti target increases. Compared with the Ti films prepared with target A, the resistivity of the Ti films prepared by targets B and C are reduced by 10.8% and 23.7%, respectively. The results indicate that the higher the purity of the Ti target, the better the electrical properties of the Ti films, which can be ascribed to the fewer impurity elements in Ti films. The oxygen element in the low-purity Ti target that exists on the surface of the film can hinder the migration of carriers, resulting in a decrease in the Hall mobility and an increase in the resistivity of the film [30]. When Ti films are used as barrier layers in integrated circuits, lower resistivity is required [31].

### 3.5. Morphology and Resistivity of Films Prepared at Different Annealing Temperatures on High-Purity Ti Targets

Figure 8 shows the metallographic structure of the Ti targets with the same purity annealed at different temperatures (target C1: without annealing treatment; target C: annealed at 700 °C; target C2: annealed at 1000 °C). The grain size of targets C1, C and C2 increases in turn [32]. There are a large number of twins and a small amount of fine equiaxed α-phase grains in target C1, and target C is composed of equiaxed α-phase grains. Different from the microstructure of targets C1 and C, the target C2 is composed of coarse β-phase and lamellar α-phase and a small amount of equiaxed α-phase [33] due to recrystallization [34].

Figure 9 shows the surface and cross-sectional morphologies of the Ti films prepared with a deposition time of 10 min. The Ti film prepared by target C1 has uneven larger particle size on the surface. The Ti film prepared by target C has a smooth surface. The surface of Ti film prepared by the target C2 has a few microcracks. The thicknesses of the Ti films prepared by three kinds of Ti are relatively uniform, which is consistent with the previous results that Ti target purity influences film thickness.

Figure 10 shows the thickness and resistivity of Ti films prepared by sputtering three high-purity Ti targets for 10 min. The square resistances of Ti films measured by a four-probe resistance tester are 44.07, 42.31 and 84.26 Ω/☐, respectively. The film thicknesses are 177.5, 163.3 and 154.3 nm, respectively. According to Formula (1), the resistivities of Ti films prepared by targets C1, C and C2 are 7.8 × 10^−6^, 6.9 × 10^−6^ and 13.0 × 10^−6^ Ω∙m, respectively. It can be seen that the thickness of the films prepared by targets C1 and C2 decreases, and resistance first decreases and then increases. The microstructure of Ti target C1 is mainly fine twins, the microstructure of target C annealed at 700 °C is equiaxed α-phase grains. After the increase in annealing temperature to 1000 °C, the microstructure of target C2 is mainly coarse β-phase. The average grain sizes of Ti targets increase with the annealing temperatures increase. The smaller the grain size, the more the grain boundaries and the higher the grain boundary density on the target surface. Due to the high-stress deformation and the existence of grain boundary energy, atoms at the grain boundary have higher energy and are easily sputtered [35]. Based on the analysis of thickness and resistivity of Ti films prepared by targets C, C1 and C2, it can be seen that Ti target C with equiaxed α-phase presents better sputtering performance.

## 4. Conclusions

Compared to the low-purity Ti target (99.7%) and the industrial Ti target (99.99%), the high-purity Ti target (99.999%) annealed at 700 °C under argon protection presents better crystallinity and more uniform equiaxed α-phase grains.Compared with films prepared by low-purity Ti target (99.7%) and industrial Ti target (99.99%), Ti films prepared by sputtering high-purity Ti target (99.999%) for 10 min have a smoother surface, evenly distributed small particles, increased and more uniform film thickness (film thickness = 163.3 nm), lower surface roughness (Ra = 121 nm), lower resistivity (6.9 × 10^−6^ Ω∙m) and better comprehensive performance. Deposition rates of Ti films prepared by high-purity Ti targets gradually decrease with an increase in sputtering time. Deposition rates of Ti film with sputtering for 10, 20 and 30 min are 16.3, 13.3 and 10.3 nm/min, respectively.Compared with the high-purity Ti target without annealing treatment and annealed at 1000 °C, Ti films prepared by the Ti target annealed at 700 °C have better overall performance.

## Figures and Tables

**Figure 1 materials-15-02661-f001:**
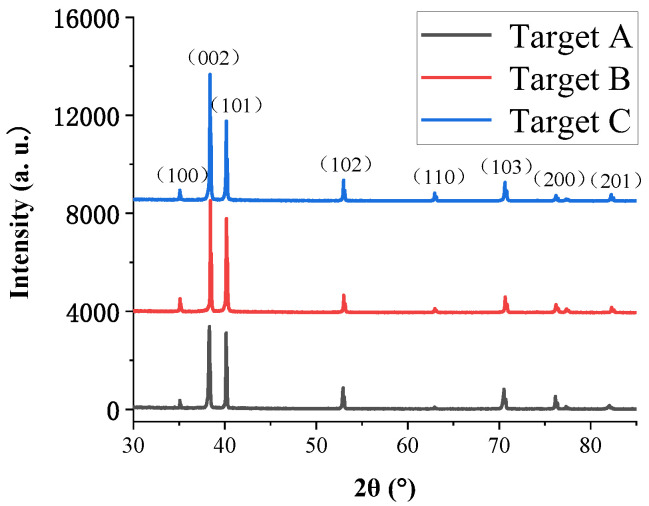
XRD patterns of Ti targets with different purities.

**Figure 2 materials-15-02661-f002:**
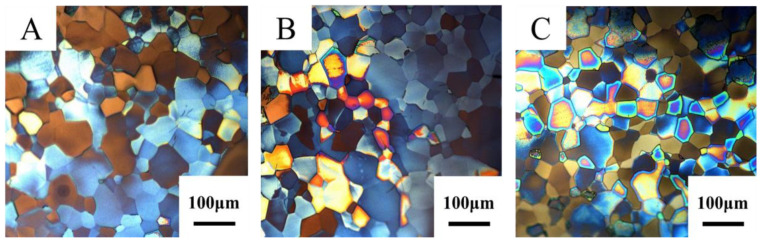
Metallographic structures of Ti targets with different purities. (**A**) Target A; (**B**) Target B; (**C**) Target C.

**Figure 3 materials-15-02661-f003:**
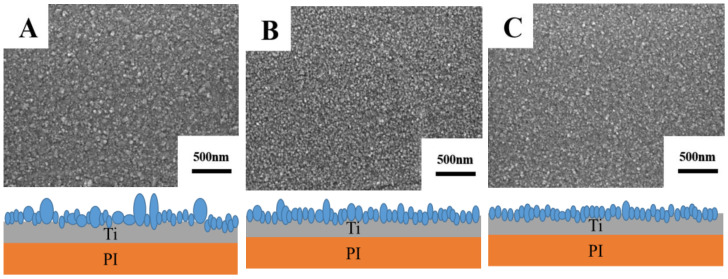
Surface morphologies of films prepared from Ti targets with different purities. (**A**) Target A; (**B**) Target B; (**C**) Target C.

**Figure 4 materials-15-02661-f004:**
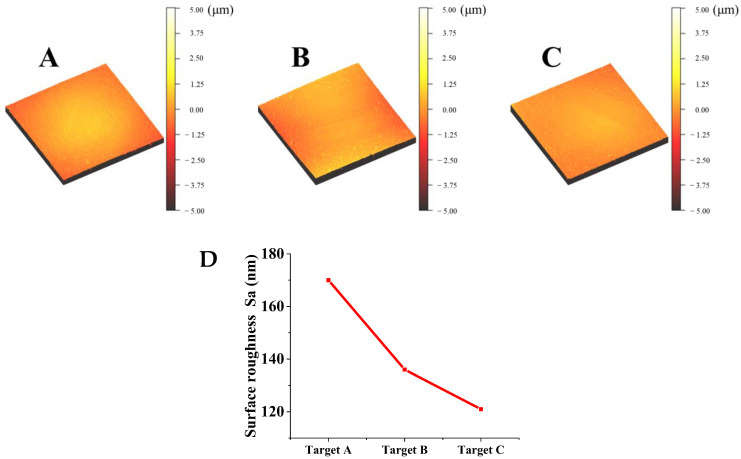
Surface roughness of films prepared from Ti targets with different purities. (**A**) Target A; (**B**) Target B; (**C**) Target C; (**D**) Variation of the surface roughness with different purities of targets.

**Figure 5 materials-15-02661-f005:**
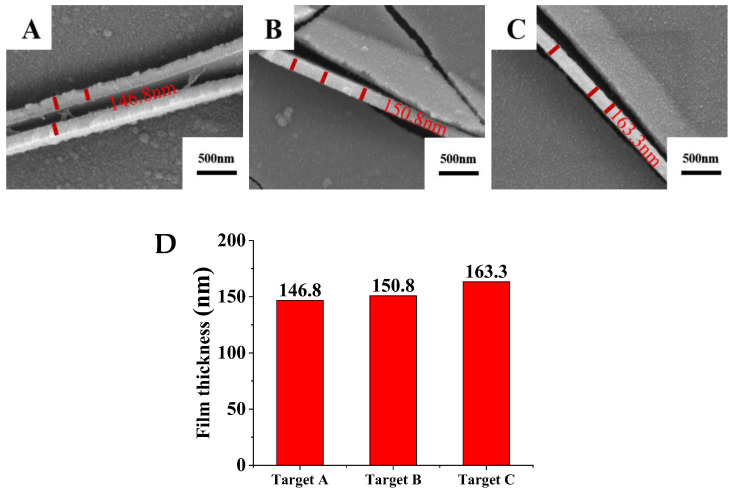
Cross-sectional morphologies and film thicknesses of films prepared from Ti targets with different purities. (**A**) Target A; (**B**) Target B; (**C**) Target C; (**D**) Variation of the film thickness with different purity of targets.

**Figure 6 materials-15-02661-f006:**
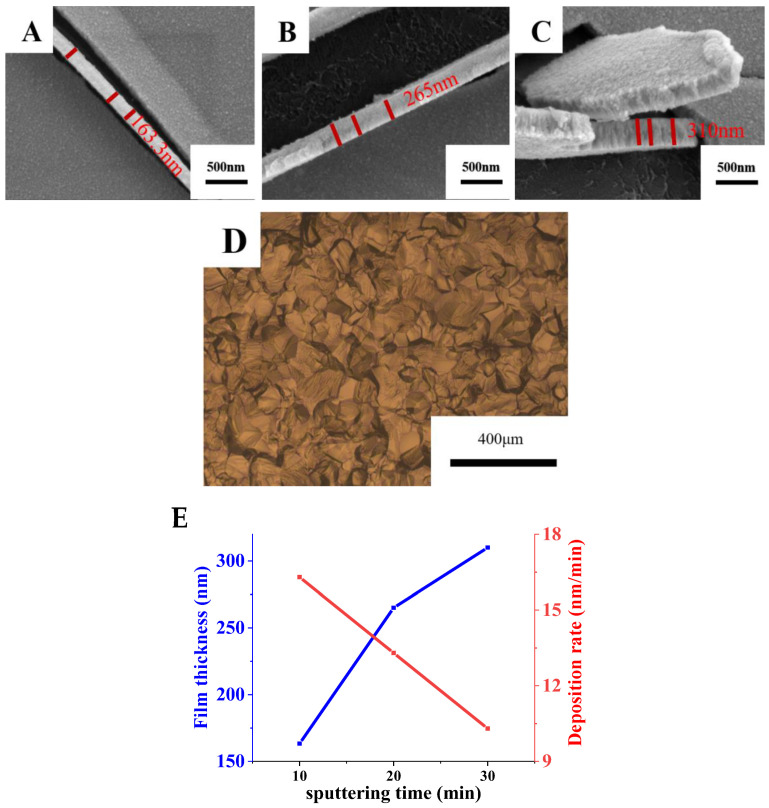
Sputtering morphology of target C and cross-sectional morphologies and deposition rates of Ti films prepared by target C with different sputtering times. (**A**) Sputtering time: 10 min; (**B**) Sputtering time: 20 min; (**C**) Sputtering time: 30 min; (**D**) Sputtering morphology of target C; (**E**) Variation of the film thickness and deposition rate with sputtering time.

**Figure 7 materials-15-02661-f007:**
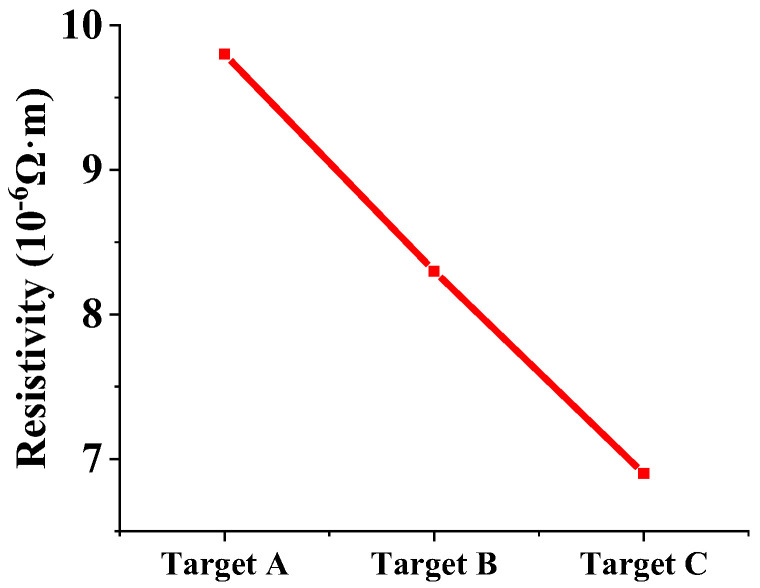
Resistivity of films prepared by Ti targets with different purities.

**Figure 8 materials-15-02661-f008:**
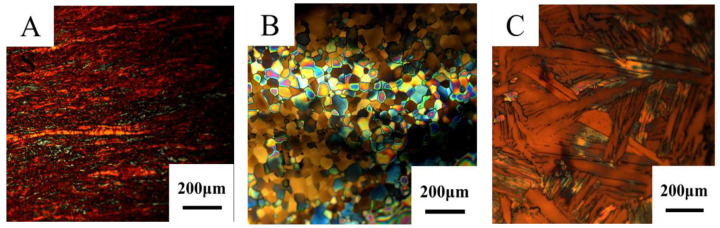
Microstructure of high-purity Ti targets at different annealing temperatures. (**A**) Target C1; (**B**) Target C; (**C**) Target C2.

**Figure 9 materials-15-02661-f009:**
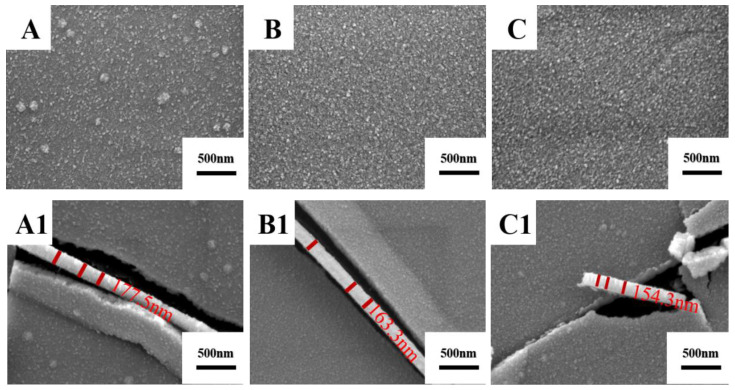
Surface and cross-sectional morphologies of films prepared by three high-purity Ti targets. (**A**) Surface morphology of film prepared by target C1; (**B**) Surface morphology of film prepared by target C; (**C**) Surface morphology of film prepared by target C2; (**A1**) Cross-sectional morphologies of film prepared by target C1; (**B1**) Cross-sectional morphologies of film prepared by target C. (**C1**) Cross-sectional morphologies of film prepared by target C2.

**Figure 10 materials-15-02661-f010:**
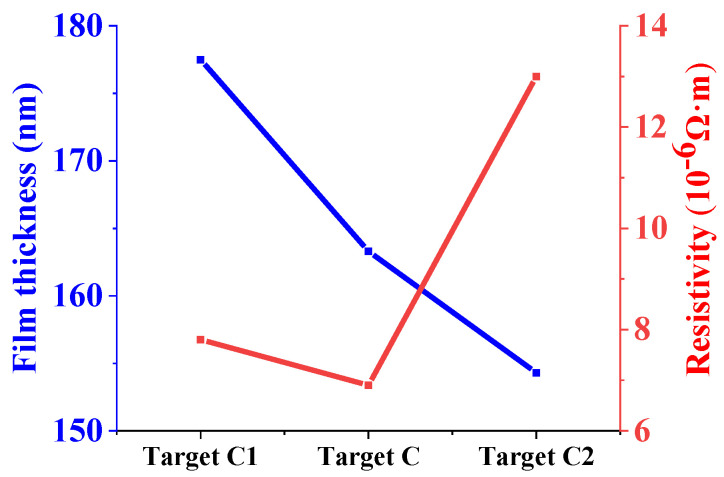
Thickness and resistivity of three kinds of films.

**Table 1 materials-15-02661-t001:** Ti targets of different purities and annealing temperatures.

Ti Targets Label	Annealing Temperature	Purity (Mass Fraction)
A	700 °C	99.7%
B	700 °C	99.99%
C	700 °C	99.999%
C1		99.999%
C2	1000 °C	99.999%

## Data Availability

The data presented in this study are available on request from the corresponding author. The data are not publicly available due to these data are part of ongoing research.

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
