# Peer review of "Effects of Ti Target Purity and Microstructure on Deposition Rate, Microstructure and Properties of Ti Films"

_materials, 2022, doi:10.3390/ma15072661_

Round 1
Reviewer 1 Report
This manuscript is devoted to the study of titanium films synthesized by magnetron sputtering. The subject area of the manuscript is interesting and deserves attention. The introduction reflects previously conducted research and defines the purpose of the study. In general, the manuscript leaves a positive impression. At the same time, the manuscript contains several shortcomings that need to be corrected before publication.
1. Please indicate the peaks in Figure 1.
2. Some figures do not contain the marking "a,b c". Provide a description of the notation in the caption to the figures.
3. The thickness of the films. Figure 3 shows that Target A has the greatest inhomogeneity in the height of the film. At the same time, Figure 5 shows that the thickness of the Target A is the smallest. It seems to me that in order to correctly represent the results, it is necessary to measure the thickness at several points. Or designate Figure 5 as "the smallest film thickness".
4. The conclusions are written very succinctly. In the conclusions, each stage of the study should be reflected in detail.
Reviewer 2 Report
This submission is devoted to study the effect purity and annealing of target on the characteristics of magnetron sputtering titanium coating. The research topic and design of study are very simple and does not have a great scientific significance. Despite the relatively high quality of the presentation of the results, the manuscript is not of great scientific soundness, significance and originality.
The scientific soundness of this manuscript can be increased if more attention is paid to the discussion of the results and comparison with the literature data. In the introduction, the authors showed that the task of obtaining high quality titanium films is very important, but the authors did not show how far they could advance in solving this problem. In my opinion, it is quite obvious that the higher the purity of the target, the better the quality of the growing film. What is the real and concrete significance of the results obtained? How can they be used in the future?
Among the specific disadvantages I would like to point out the following:
- Why was polyimide used as a substrate? Why was only one substrate used? The choice of substrate is very important in this case, because the substrate greatly affects the structure of the coating.
- Section 2 (Test methods) should be titled Materials and Methods. Moreover this section is too short. It is necessary to give a more detailed description of the methodology for study the samples. Scan area for topography study, types of detectors for SEM measurements, 2 teta range and radiation type for XRD, parameters of sheet resistance measurements etc.
- What is the elemental composition of impurities in targets?
- XRD data are discussed in terms of intensity of peaks. But what about peak width? I would suggest estimating grain size based on Scherrer equation or Rietveld method and compare results with microstructure data.
- Figure 3. On what basis were the figures in the lower part drawn up? Is this a model representation or a hypothetical one?
- Figure 4. Surface reconstructions are not informative. Color scales need to be introduced. How was the Sa roughness calculated? There are many parameters that describe the surface topography. Why was only Sa chosen? What were the errors of Sa calculation? How many scans were used to estimate Sa?
- Figure 5. The thicknesses of the films are close. What were the errors of thickness measurements?
- Why are XRD data presented only for A,B, C samples and no data for C1, C2.
- Only the resistivity of C, C1, C2 were shown. What about A and B samples. Also the four-probe method allows calculating the carrier concentration and its mobility. This data is very important for discussion of effect the structure and impurities of films on electrical properties.
- The manuscript is sorely lacking in discussion and explanation of the results. For example, what are the mechanisms of influence of target annealing on the growth of films and their characteristics?
Nevertheless, despite the dubious scientific significance, the study is well structured, described and written, the conclusions are confirmed by the results. I would suggest reconsidering the manuscript after major revision.
Reviewer 3 Report
The authors have measured the effect of target purity and annealing conditions on the sputtering behaviour and properties of Ti films and shown that purity/crystallinity affects them. However, they have not addressed the direct effect of the contamination on the films and sputtering behaviour. It is important to know what the contaminants are to understand what is happening. For example, if there is O contamination, the presence of Ti oxides in the target will change the sputtering yield and affect the deposition rate. It will also affect the mobility of the Ti atoms on the substrate which will affect the smoothness regardless of the crystallinity of the target. If the impurities are metallic, there may be totally different effects. Therefore is important to analyse the impurities in the target and in the deposited films.
In lines 145-146, it is stated “The impurity elements in the Ti film can promote to form particles on the films’ surface and hinder Ti atoms diffusion, thereby reducing film deposition rate [24].” The impurities in the surface of the films cannot affect the deposition rate – that is determined by the amount of material being sputtered from the target. Also, reference [24} is not relevant here. It refer to the cold spraying of Cu thick films to form a sputtering target, not the effects of sputtering on the deposition rate on a substrate. Also, in the introduction, reference [9] shows how sputter yield can change because of morphology differences in the target, There has been no attempt to relate this analysis to the results here which might point out the effects of target crystallinity vs. target impurity.
It is also important to explain why there is a decrease in deposition rate with sputtering time. Is this due to Ti gettering of the sputtering gas thus reducing the background pressure, or the effects detailed in ref [9], for example?
The reduction in resistivity as the target purity is increased (fig 7) is normal, especially if the impurities are oxides. Fig 10 shows the change of resistivity depending on target annealing. Is this due to change in crystal structure of the films or due to different levels of contamination?
The Introduction includes references to reactive sputtering of TiAlN and in a Ar + N2 atmosphere and the resistivity Ti/Ag films, neither of which is relevant to this paper.
Round 2
Reviewer 2 Report
Dear Authors,
Thank you for your responce!
I do not agree with all the points, but your answers are well reasoned. Therefore, I can accept them.
However, there are a couple of points that still need to be improved.
1) Detailed information about the composition of the target should be shown either in the text of the article or in an supplementary file.
2) You have not made any changes in figure 4 ABC. From the figure it is absolutely not clear what the changes in color on the topography reconstructions mean. Need to show color gradient (color scales) and sizes of scan areas.
Reviewer 3 Report
The authors have, in their cover letter, shown a table of impurity measurements for the different targets which they refer to in the text (lines 136-136). They should include the table either as an appendix or as supplementary information.
Line 18 They have quoted a deposition rate of 16.63 nm/min. This is exceptional accuracy for a process with inherent variability. It should probably be given as 16,6 nm/min. They should also check the other figures in the text to make sure the resolution of the values given is reasonable.
Line 25 “Titanium (Ti) were widely…” should be “Titanium (Ti) is widely…”
Line 29 “….is closely…”
Line 32 “and expectation” should be “the expectation”
Line 149 “large particles exsited on the surface….”. Delete “exsited”, it is not necessary (and spelled wrongly).
Lines 172-173 “The reason of deposition rate decrease with extension of time is that the atoms inside the target need higher energy to be sputtered out.” Do they mean that because grain boundary regions are more easily sputtered the sputter yield is decreasing with time because the target surface becomes enriched with larger grains? This needs a clearer explanation.
Lines 173-174 The sentence “In addition, with the extension of sputtering time, more and more plasma collide with Ar + in the vacuum chamber, which reduces the sputtering efficiency [28].” This does not make sense I do not understand the argument at all. Please clarify.
Lines 205-206 Specify the different annealing temperatures.
Line 232 “C annealed at is equiaxed” Temperature is missing
Lines 238-240 “Compared with high-purity target C, targets A and B contain more impurity elements which can reduce the atoms diffusion and result in the resistivity under the same sputtering conditions [30] .” This sentence does not make sense. There is something missing from it. In any case it, does not appear relevant here.
References: several of the references have incomplete information, for example, 25-28. They should be thoroughly checked.
Language: the text should be thoroughly checked for correct English. In general the text is understandable but some revision would increase the clarity.
